# Translation repression via modulation of the cytoplasmic poly(A)-binding protein in the inflammatory response

**Xu Zhang[1†], Xiaoli Chen[2†], Qiuying Liu[1], Shaojie Zhang[2], Wenqian Hu[1]\***

[1]Department of Biochemistry and Molecular Biology, Mayo Clinic, Rochester, United States; [2]Department of Computer Science, University of Central Florida, Orlando, United States

**Abstract** Gene expression is precisely regulated during the inflammatory response to control infection and limit the detrimental effects of inflammation. Here, we profiled global mRNA translation dynamics in the mouse primary macrophage-mediated inflammatory response and identified hundreds of differentially translated mRNAs. These mRNAs' 3'UTRs have enriched binding motifs for several RNA-binding proteins, which implies extensive translational regulatory networks. We characterized one such protein, Zfp36, as a translation repressor. Using primary macrophages from a Zfp36-V5 epitope tagged knock-in mouse generated by CRISPR/Cas9-mediated genome editing, we found that the endogenous Zfp36 directly interacts with the cytoplasmic poly(A)-binding protein. Importantly, this interaction is required for the translational repression of Zfp36's target mRNAs in resolving inflammation. Altogether, these results uncovered critical roles of translational regulations in controlling appropriate gene expression during the inflammatory response and revealed a new biologically relevant molecular mechanism of translational repression via modulating the cytoplasmic poly(A)-binding protein.

**\*For correspondence:** hu.
wenqian@mayo.edu

[†]These authors contributed
equally to this work

**Competing interests:** The
authors declare that no
competing interests exist.

**Reviewing editor:** Torben Heick
Jensen, Aarhus University,
Denmark

## Introduction

Precise generation of gene products is essential for cells to mount proper responses to physiological and pathological stimuli. This accuracy is achieved by multiple layers of regulations that occur synchronously at different steps of gene expression, including transcription, splicing, translation, mRNA stability, and protein modification and degradation. Importantly, malfunction of these regulatory circuits contributes to many pathological conditions. Thus, characterizing the regulatory networks controlling gene expression is of both biological and clinical significance.

Macrophages are an important type of innate immune cell involved in both detecting exogenous and endogenous danger signals and initiating proper immune responses (reviewed in *Medzhitov and Janeway, 2000*). These danger signals are recognized as pathogen-associated molecular patterns (PAMPs) by pattern-recognition receptors (PRRs). PAMPs are molecules that are absent in host cells but broadly shared by pathogens. For example, macrophages can detect bacterial infection by recognizing bacterial lipopolysaccharide (LPS), a cell membrane component present in gram-negative bacteria, by a PRR, toll-like receptor 4. This recognition activates signaling transduction cascades that result in rapid expression of inflammatory response genes to control infection. Many gene products from the inflammatory response, such as the proinflammatory cytokine TNF (tumor necrosis factor alpha), however, are also toxic to healthy tissues (reviewed in *Kalliolias and Ivashkiv, 2016*). Thus, the expression of these inflammatory genes is tightly controlled by multiple regulatory circuits to maintain a delicate balance between managing infection/injury and damaging normal tissues (reviewed in *Hanada and Yoshimura, 2002*). Critically, malfunction of these

**eLife digest** DNA sequences called genes produce RNA molecules, some of which (the "messenger RNAs") go on to be 'translated' to make proteins. This gene activity enables cells to react to their surroundings. For example, immune cells called macrophages produce hundreds of RNA molecules and proteins as part of an inflammatory response that defends the body against an infection. However, many of these molecules can also damage healthy tissue, so many layers of regulation control when, and how much of, these molecules are made.

There are several ways to control how many proteins a cell produces. For example, cells might regulate how many messenger RNA molecules (also called mRNAs for short) are produced from a gene, or control how many proteins are translated from those mRNA molecules. Previous studies of how inflammatory responses are regulated have largely focussed on how mRNA production is controlled. Much less is known about the role that regulating mRNA translation has on the inflammatory response.

By studying mouse macrophages, Zhang, Chen et al. have now identified hundreds of proteins whose production is regulated during an inflammatory response by controlling their translation from mRNA molecules. A group of RNA-binding proteins produced by the macrophages perform this regulation. Further observation revealed that a particular RNA-binding protein called Zfp36 prevents the translation of several important mRNAs, thereby helping to end an inflammatory response.

Zhang, Chen et al. then genetically engineered mice to produce a version of Zfp36 that has a 'tag' attached to it that makes the protein easier to detect. Studying the activity of Zfp36 in these mice revealed that this RNA-binding protein works by interacting with another protein that normally binds to structures known as poly(A) tails at the end of the mRNA molecules.

Zhang, Chen et al. believe that similar genetic engineering approaches could help researchers to study how other RNA-binding proteins work in living animals. In addition, by better understanding how inflammatory responses are regulated it may be possible to investigate new ways of treating conditions where this response is prolonged, such as in autoimmune disorders like rheumatoid arthritis.

regulations can result in pathologic inflammation that is associated with a wide variety of human diseases, including obesity, cardiovascular and neurodegenerative diseases, and cancer (Reviewed in *Kotas and Medzhitov, 2015*). Thus, characterizing regulatory programs in the inflammatory response will both reveal fundamental mechanisms of gene expression and provide insights into inflammation-associated diseases.

Previous studies have identified multiple programs controlling mRNA production and degradation in macrophage-mediated inflammatory responses (reviewed in *Carpenter et al., 2014*; *Medzhitov and Horng, 2009*). For example, stimulation of PRRs by PAMPs can activate several transcription factors, such as nuclear factor-κB, interferon-regulatory factors, and CCAAT/enhancer-binding protein-δ, which results in the rapid induction of inflammatory response genes. These genes include both cytokines (e.g. TNF) to control infection and negative regulators of inflammatory signaling pathways (e.g. suppressors of cytokine signaling proteins) to limit the detrimental effects of inflammation. Besides transcriptional regulations, inflammatory response genes are also subject to controls at the mRNA stability level. The 3'-untranslated regions (UTRs) of many inflammatory cytokine mRNAs have sequence motifs, such as AU-rich elements (AREs) and constitutive decay elements (CDE) that can promote rapid mRNA degradation (*Garneau et al., 2007*; *Stoecklin et al., 2003*). These mRNA decay mechanisms help to prevent sustained expression of potentially harmful cytokines, thereby contributing to resolving inflammation. Together, these regulations at the mRNA production and degradation levels ensure appropriate transcriptomic responses during inflammation.

In eukaryotic cells, mRNA degradation is intimately linked with mRNA translation (reviewed in *Roy and Jacobson, 2013*). Interestingly, temporal regulation of translation and mRNA decay is an important mechanism of inhibiting gene expression. For example, certain microRNAs down-regulate the expression of their target mRNAs by first inhibiting translation and then promoting mRNA degradation (*Bazzini et al., 2012*; *Djuranovic et al., 2012*). Compared to our knowledge of mRNA

stability regulations, our understanding of mRNA translational control during the inflammatory responses, however, is still very limited. Recent observations from activated macrophage-like cell lines indicate that several mRNAs are differentially translated (*Schott et al., 2014*), which implies important roles of controlling mRNA translation during the inflammatory response. The *trans*-acting factors regulating the translation of these mRNAs, however, are still largely unknown.

In this study, we explored translational regulation of the inflammatory response mediated by mouse primary bone-marrow-derived-macrophages (BMDMs). Using ribosome profiling, we identified hundreds of differentially translated mRNAs. Interestingly, the 3'UTRs of these mRNAs have significantly enriched binding motifs for several RNA-binding proteins (RBPs) expressed in the activated BMDMs, which suggests widespread mRNA translational regulatory networks. We functionally characterized one such RBP, Zfp36, which is required for resolving inflammation, as a translational repressor. Using primary BMDMs from a V5-epitope tag knock-in mouse generated by CRISPR/Cas9-mediated genome editing, we identified the target mRNAs of the endogenous Zfp36 by CLIP-seq (cross-linking immunoprecipitation followed by high-throughput sequencing). Moreover, using quantitative proteomics, we found that the endogenous Zfp36 predominantly interacts with the cytoplasmic poly(A)-binding protein (Pabpc1) in an RNA-independent manner in primary activated BMDMs. Critically, this interaction is required for the translational repression of the Zfp36 target mRNAs, including the mRNAs encoding several important proinflammatory cytokines, in the activated BMDMs. Collectively, these results highlight critical roles of translational regulations in controlling appropriate gene expression during the inflammatory response and reveal a new biologically relevant molecular mechanism of translational repression via modulating Pabpc1.

## Results

### Global translational profiling during the macrophage-mediated inflammatory response

We used the response of mouse primary BMDMs to LPS as a model to study translational regulations in inflammation, because this is one of the best-characterized inflammatory responses (reviewed in *Medzhitov and Janeway, 2000*). Specifically, mouse primary BMDMs were treated with 100 ng/ml LPS and collected at 0, 1, 2, 4, and 6 hr after LPS stimulation. Cells harvested at each time point were split into two fractions, which were subjected to either RNA-seq or ribosome profiling (Ribo-seq) (*Figure 1A*). In RNA-seq, ribosomal-RNA-depleted total RNAs were randomly fragmented, followed by directional cloning for sequencing. In Ribo-seq, the ribosome-protected mRNA fragments (RPFs) were isolated by sucrose-density gradients and then subjected to directional cloning and sequencing. The resulting reads were then aligned to a well-annotated mouse transcriptome, with more than 300 million reads mapped in total. Thereby, we profiled mRNA expression and the ribosome-mRNA association in parallel during the BMDM-mediated inflammatory response.

Our Ribo-seq datasets were highly reproducible, as indicated by comparing the results from two biological replica sets of BMDMs (*Figure 1—figure supplement 1A*). Moreover, the following fundamental features of translation can be captured at single-nucleotide resolution from these data. First, the RPFs precisely delineate known coding sequences (CDSs) and their exons, with 12-nt and 15-nt offsets upstream of the translation start codons and termination codons, reflecting known distances from RFP 5'-termini to the P- and A- site codons, respectively (*Figure 1B*). Second, the RPFs have 3-nt codon periodicity, a key feature of translocating ribosomes, but the RNA-seq reads do not (*Figure 1C*). Third, the RPFs are highly specific to CDS regions compared with the RNA-seq reads (*Figure 1D*). Collectively, these observations indicate good data quality for global analysis of translational regulation in the inflammatory response.

To identify differentially translated mRNAs in the inflammatory response, we calculated the apparent translation efficiency (TE) for each mRNA expressed above a threshold (FPKM $\geq$10) at each time point. Here, we defined the apparent TE as the enrichment of RPFs over RNA-seq reads in the CDS regions. We found 724 mRNAs with significant (empirical p<0.05) TE changes across the five time-points during the LPS treatment (*Figure 1E*) (*Supplementary file 1*). Gene ontology analysis revealed that these mRNAs are involved in diverse pathways (*Figure 1—figure supplement 1B*), including the TNF and NF-kB signaling pathways that are critical in inflammatory responses. Indeed, *TNF* mRNA, encoding the essential inflammatory cytokine TNF, is translationally regulated. This

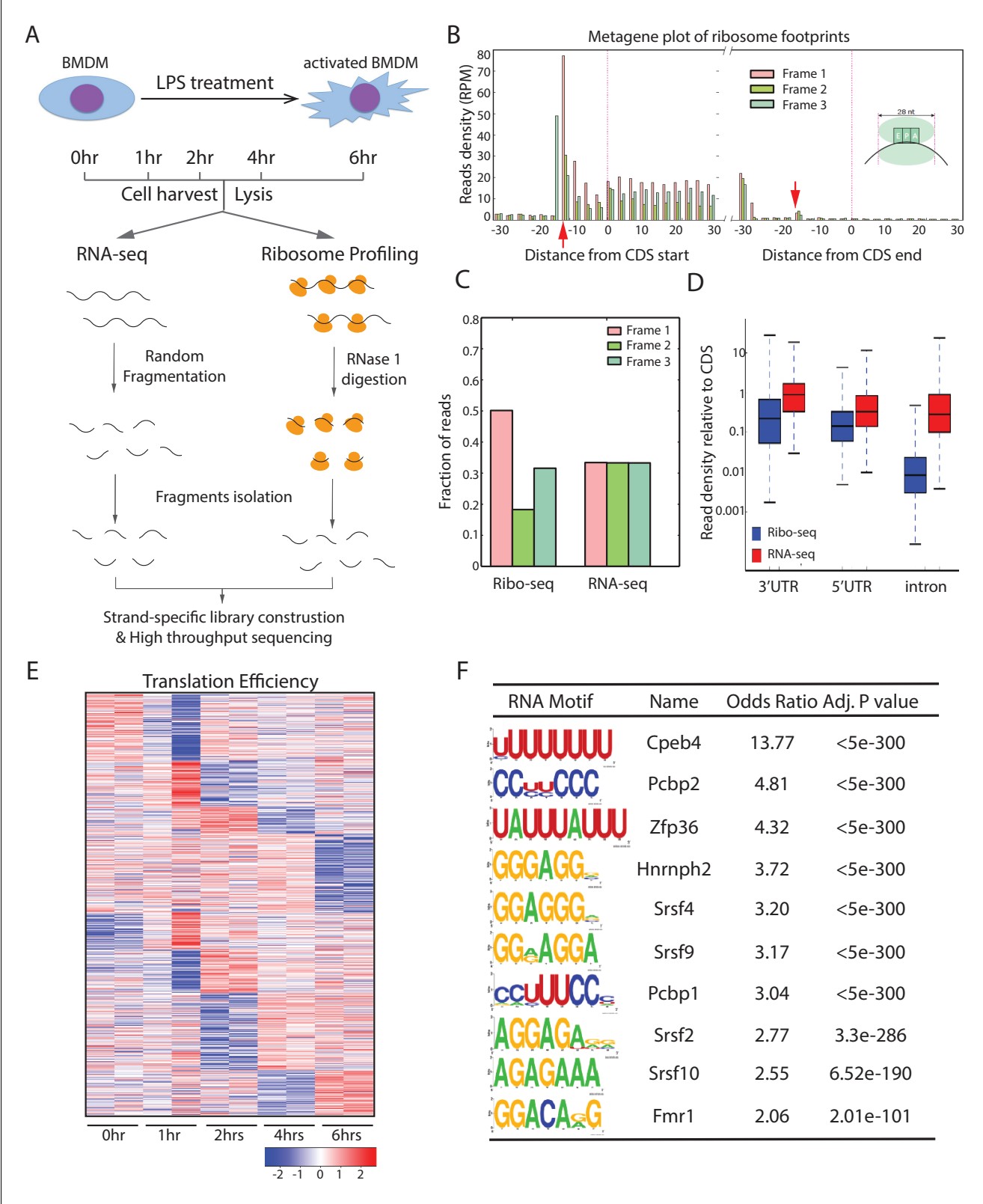

**Figure 1.** Global translation profiling of mouse primary BMDM-mediated inflammatory response. (**A**) Workflow of parallel ribosome and RNA profiling during the BMDM-mediated inflammatory response. (**B**) Metagene plots show the rise and fall in 28-nt ribosome footprints (RFPs) density (reads per million uniquely mapped reads, RPM) near starts and stops of annotated CDS, respectively. The 12-nt and 15-nt offsets (indicated by the red arrows) from starts and stops reflect distances from RFP 5′ termini to the ribosome P- and A-site codons at translation initiation and termination, respectively.

*Figure 1 continued on next page*

*Figure 1 continued*

(C) Subcodon resolution of ribosome footprints. Note that 3-nt codon periodicity relative to the known CDSs is seen for 28-nt RFPs but not RNA-seq reads. (D) Ribosome footprints are highly specific to coding regions. Boxplots show density of Ribo-seq and RNA-seq reads at 5'UTRs, 3'UTRs, and introns relative to that of the associated CDSs. (E) Clustering analysis of the translation efficiency (TE) of the 724 genes differentially translated (empirical p<0.05) in the inflammatory response. Heatmap displays mean row-centered log2 TE values at 0, 1, 2, 4, 6 hr post LPS treatment. (F) RNA-binding protein motifs enriched within the 3'UTRs of the 724 translationally regulated genes. The 10 most enriched motifs of macrophage-expressed RBPs are shown along with enrichment statistics.

The following figure supplements are available for figure 1:

**Figure supplement 1.** Reproducibility of Ribo-seq and GO analysis of differentially translated mRNAs in the inflammatory response.

**Figure supplement 2.** TNF mRNA is translationally repressed during late stages of inflammatory response.

**Figure supplement 3.** Verification of translational repression of TNF mRNA at late time points in LPS-stimulated BMDMs.

**Figure supplement 4.** Zfp36 is a translational repressor.

mRNA is transcriptionally induced from 0 to 1 hr upon LPS stimulation and then decreases from 1 to 2 hr, and the mRNA level stays relatively stable untill 6 hr post-stimulation. The RPFs of *TNF* mRNA, however, decreases significantly from 1 to 6 hr, with few *TNF* RPFs at 6 hr (*Figure 1—figure supplement 2*), which indicates *TNF* mRNA is translationally repressed in the late stages of inflammation. Indeed, the apparent TE of *TNF* mRNA in the BMDMs had a significant decrease from 1 hr to 6 hr post LPS stimulation (*Figure 1—figure supplement 3A*). To further confirm this result, we performed polysome analysis followed by qRT-PCR to monitor the distribution *TNF* mRNA across the sucrose gradient (*Figure 1—figure supplement 3B*). Compared with *TNF* mRNA at the 1 hr time point, the *TNF* mRNA had a significant shift from the polyribosome region to the mRNP region on the sucrose gradient at 6 hr post the LPS treatment (*Figure 1—figure supplement 3B and C*), indicating that this transcript was translationally repressed. Importantly, the distribution of a control mRNA, *Gapdh* mRNA, did not change between the 1 hr and the 6 hr time points (*Figure 1—figure supplement 3C*), which indicates the specificity of the translational repression on the *TNF* mRNA. Collectively, these observations indicated that a large number of mRNAs are regulated at the translational level during the inflammatory response.

## Identification of potential translational regulators in the activated BMDMs

Translation of mRNA can be modulated by cis-elements in UTRs and trans-acting factors, such as RBPs and microRNAs. To identify the potential translational regulators in the inflammatory response, we performed the following computational analysis. First, we searched the 3'UTRs of the 724 differentially translated mRNAs for significantly enriched motifs (adjusted $p<10^{-100}$, Fisher's exact test) that match with the known binding sites of ~200 RBPs identified through in vitro binding studies (*Ray et al., 2013*), which generated a list of RBPs that can bind these mRNAs. Next, from this list of RBPs, we chose those that are expressed (FPKM $\geq$10) in BMDMs, which resulted in a group of RBPs (*Figure 1F*) that may be potential translational regulators in the inflammatory response. Among these RBPs are several known translational regulators, such as Pcbp1, Pcbp2, and Cpeb4 (*Hu et al., 2014*; *Makeyev and Liebhaber, 2002*), which supports the validity of this approach.

Here, we focused on Zfp36 (TTP), an ARE(AU-rich element)-binding protein, for several reasons. First, genetic studies in mouse indicated that Zfp36 is required to resolve inflammation (*Taylor et al., 1996*). Second, Zfp36 is abundantly expressed in activated BMDMs (FPKM >200). Third, although Zfp36 was characterized to promote target mRNA degradation (reviewed in *Brooks and Blackshear, 2013*), its role in mRNA translational control is not well understood in primary BMDMs.

To test whether Zfp36 can regulate mRNA translation, we performed the tethering experiments, which allow us to determine the function of a target RBP without knowing its substrate mRNAs (*Coller and Wickens, 2007*). Specifically, using the robust and specific interaction between the

bacteriophage λN polypeptide and the BoxB RNA motif, we tethered a λN-Zfp36 fusion protein to the 3'UTR of a firefly luciferase (FLuc) mRNA (*Figure 1—figure supplement 4*). Luciferase activity and the *FLuc* mRNA level were then assayed in 293 T cells. Tethering Zfp36 reduced both the luciferase activity and the *Fluc* mRNA level (*Figure 1—figure supplement 4*). Importantly, the translatability of the *Fluc* mRNA, defined as the luciferase activity normalized to the *Fluc* mRNA level, was significantly decreased in a Zfp36-dependent manner (*Figure 1—figure supplement 4*), which indicated that Zfp36 can repress mRNA translation. This result is consistent with previous observations in the Zfp36 knockout BMDMs that the *TNF* mRNA, a Zfp36 target, increases ~two-fold compared with that in wild-type BMDMs; but the TNF protein increased ~five-fold (reviewed in *Taylor et al., 1996*), which indicates that Zfp36 also represses target mRNA translation in primary BMDMs.

Next, we aimed to determine how Zpf36 represses mRNA translation in primary BMDMs and to identify the mRNA targets of endogenous Zfp36.

## A V5-epitope tag knock-in mouse for studying the endogenous Zfp36

Previous studies using immunoprecipitation (IP) of overexpressed Zfp36 or the yeast two-hybrid system have identified many proteins with which Zpf36 can interact (reviewed in *Brooks and Blackshear, 2013*). Although useful mechanistic insights can be obtained, there are three caveats in interpreting these previous results: (1) cell lines may not represent the corresponding primary cells; (2) proteins binding to the overexpressed Zfp36 may not interact with the endogenous Zfp36; and (3) for many identified interactions, their effect on gene expression are still unknown. Similar limitations also apply to the mRNA targets identified in Zpf36-overexpressing macrophage-like and non-macrophage cell lines (*Mukherjee et al., 2014*; *Tiedje et al., 2016*). One significant technical challenge of studying the endogenous Zpf36, however, is the unavailability of IP-grade antibodies for specific and efficient Zfp36 isolation.

To characterize the endogenous Zfp36-mediated regulation of gene expression in a biologically relevant setting and to overcome the technical obstacles, we created a V5-epiptope tag knock-in mouse using CRISPR/Cas9-mediated genome editing (*Figure 2A*). Specifically, a sgRNA targeting a genomic site near the stop codon region of the Zfp36 locus was coinjected into mouse zygotes with the Cas9 enzyme and an oligonucleotide containing a 51-nt V5-epitotpe tag with a ~ 60 nt homologous arm on each side of the V5 sequence. The double-stranded DNA break generated by the Cas9 facilitated in-frame knock-in of the V5-epitope tag via homologous recombination. The genetically modified zygotes were transferred into foster mouse mothers, and the Zfp36-V5 knock-in allele in the resulting mice was monitored by PCR (*Figure 2B*). We genetically purified this Zfp36-V5 knock-in allele via backcrosses. The homozygous Zfp36-V5 mice were born in the predicted Mendelian ratio from heterozygous matings (*Figure 2—figure supplement 1A*). Moreover, unlike Zfp36-deficient BMDMs (*Taylor et al., 1996*), BMDMs from the Zfp36-V5 mice have the same response to LPS as those from wild-type mice in terms of inflammatory cytokine induction and Zfp36 expression (*Figure 2—figure supplement 1B*), which indicates that the Zfp36-V5 in the knock-in mice is expressed at the same endogenous level as Zfp36 is in wild-type mice. Furthermore, the Zfp36-V5 mice are phenotypically indistinguishable from the wild-type mice. These observations are consistent with the notion that the 51-nt V5-tag knock-in sequence neither alters the Zfp36 promoter nor changes the regulatory elements in the 3'UTR.

The V5-tag allows precise detection and efficient isolation of endogenous Zfp36. Using a V5 antibody, we could unambiguously detect the endogenous Zfp36 in the BMDMs from the Zfp36-V5 mice (*Figure 2C*, *Figure 2—figure supplement 1C*). Importantly, unlike a Zfp36 antibody, the V5 antibody resulted in no nonspecific bands (*Figure 2—figure supplement 1C*), which indicates the high specificity of Zfp36 detection. Furthermore, using the V5 antibody, we could also efficiently isolate endogenous Zfp36 from UV-crosslinked (see below) LPS-treated BMDMs, as indicated by the distribution of Zfp36 in the input, IP, and supernatant samples (*Figure 2D*). Finally, the Zfp36-V5 mice provide an almost unlimited source of primary cells (that is, BMDMs). Collectively, these advantages make the Zfp36-V5 mouse a valuable tool for mechanistic characterization of endogenous Zfp36 in biologically relevant settings.

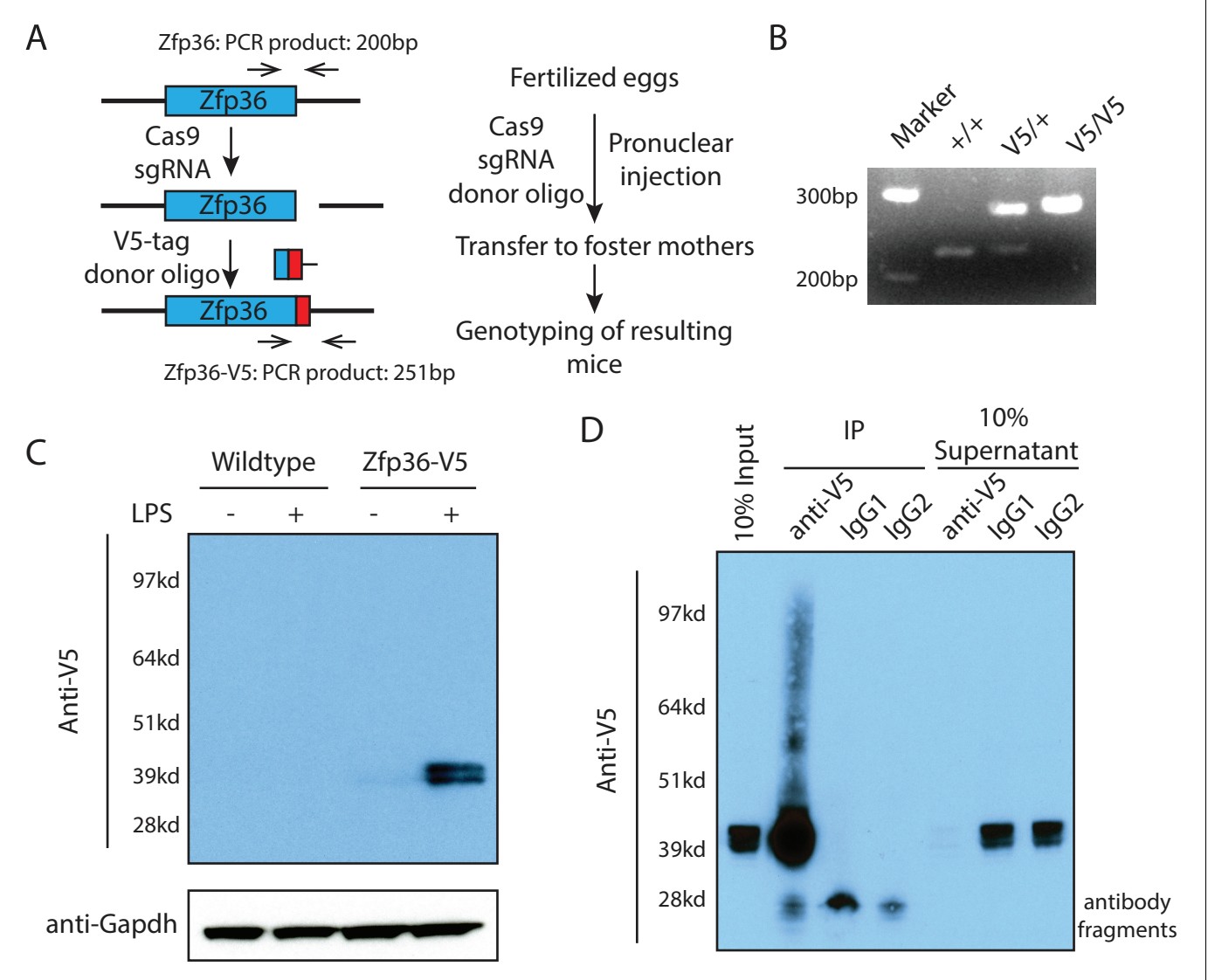

**Figure 2.** Generation of a Zfp36 V5-epitope tag knock-in mouse for mechanistic studies on the endogenous Zfp36. (**A**) Workflow for generating a Zfp36-V5 knock-in mouse via CRISPR/Cas9-mediated genome editing. (**B**) Genotyping of the Zfp36-V5 knock-in mice using the two primers shown in (**A**). (**C**) Specific and unambiguous detection of Zfp36 using the V5-epitope in BMDMs from the Zfp36-V5 mice. BMDMs from wild-type and Zfp36-V5 mice were treated with 100 ng/ml LPS for 4 hr, followed by Western blot using an anti-V5 antibody. The doublet bands of Zfp36 are due to its post-translational modifications. (**D**) Efficient isolation of endogenous Zfp36 using the V5-epitope in BMDMs from the Zfp36-V5 mice. BMDMs from the Zfp36-V5 mice were stimulated with 100 ng/ml LPS for 4 hr, followed by UV254 crosslinking. An anti-V5 antibody and two different IgGs were used to IP the endogenous Zfp36 from the cell lysates, respectively. The input, IP samples, and supernatants were subjected to SDS-PAGE and Western blot analysis using an anti-V5 antibody. The following figure supplement is available for *Figure 2*.

The following figure supplement is available for figure 2:

**Figure supplement 1.** The Zfp36-V5 mouse is normal.

## Zfp36 directly interacts with the cytoplasmic poly(A)-binding protein in activated BMDMs

To determine how Zfp36 regulates gene expression in primary BMDMs, we first identified the proteins that the endogenous Zfp36 interacts with by using IP followed by quantitative proteomics. Specifically, BMDMs from the Zfp36-V5 mice were harvested after 4 hr of LPS stimulation, when Zfp36-V5 is abundantly expressed (*Figure 2—figure supplement 1C*). The Zfp36 and its associated

proteins were IPed using the V5 antibody or IgG as control. Due to Zfp36's RNA-binding ability, the IP was performed with RNaseA, which disrupts RNA-mediated interactions, so that only protein-protein interactions can be isolated. The IPed proteins were subject to tandem mass-tag (TMT)-based quantitative proteomic analysis (*Thompson et al., 2003*) to identify the proteins specifically associated with Zfp36 (*Figure 3A*). In this method, the proteins isolated by the V5 antibody or IgG were digested by trypsin. The resulting peptides from each sample were labeled with a different isobaric mass tag, followed by pooling together for mass spectrometry (isobaric tags for relative and absolute quantification, iTRAQ) identification and quantification. The unique mass tag of each sample enables comparison of the abundance of the identified proteins in the V5 IP versus the IgG control samples. Using this approach, we identified several proteins that are significantly enriched (>2 fold in all the three biological replicas) in the V5 IP samples (*Supplementary file 2*). As the positive control, Zfp36 itself had greater than 20-fold enrichment, which indicates the validity of this method. The next-most enriched (>4 fold) and abundantly detected protein was Pabpc1, followed by 14-3-3θ (*Figure 3B*). Hereafter, we focused on Pabpc1, the cytoplasmic poly(A) binding protein 1, because of its roles in controlling mRNA translation and stability (*Mangus et al., 2003*).

To verify the RNA-independent interaction between Zfp36 and Pabpc1, we performed two additional experiments. First, we performed IPs in stimulated BMDMs from the Zfp36-V5 mice and wild-type mice with or without RNaseA using the V5 antibody. Endogenous Pabpc1 was detected by Western blot in the IP samples from the Zfp36-V5 BMDMs in an RNaseA-insensitive manner but not in that from the wild-type BMDMs (*Figure 3C*). Importantly, Gapdh, an abundant cytoplasmic protein, was not detected in the IP samples, which indicates the specificity of the IPs (*Figure 3C*). This result showed that the endogenous Zfp36-Pabpc1 interaction is not mediated by RNA. To further confirm that the interaction between these two RBPs is RNA-independent and to rule out the possibility that this interaction is mediated by the V5 tag, we performed IPs in 293 T cells expressing an HA-tagged Pabpc1 with either a FLAG-tagged Zfp36 or a FLAG-tagged RNA-binding-deficient Zfp36 mutant (Zfp36[F118N]) (*Lai et al., 2002*). We found that the HA-tagged Pabpc1 was specifically co-IPed with both the wild-type and the mutant Zfp36 in the presence of RNaseA (*Figure 3D*). To further rule out the possibility that the Zfp36-Pacbpc1 interaction is mediated by poly(A) sequences that RNaseA cannot degrade, we performed the IP experiment with RNAse1, which degrades RNA in a non-sequence-specific manner. We found that Zfp36 still interacted with Pabpc1 under this condition (*Figure 3—figure supplement 1*). Collectively, these experiments revealed that the Zfp36-Pabpc1 interaction is truly RNA-independent. Since Pabp1c is the most abundant Zfp36-associated protein identified by the quantitative proteomics, it argues that the endogenous Zfp36-Pabpc1 interaction is a direct interaction in the activated BMDMs.

To define a minimal region on Zfp36 that can specifically interact with Pabpc1, we performed structure-function mappings. In detail, a series of FLAG-tagged Zfp36 truncations (*Figure 3E*) were coexpressed in 293 T cells with an HA-tagged Pabpc1 for co-IP, respectively. We took two steps to ensure that the identified interaction is RNA-independent. First, all the Zfp36 truncations had the F118N mutation (Zfp36-m), which abolishes RNA-binding ability (*Lai et al., 2002*). Second, the IPs were performed with RNaseA. This analysis resulted in a minimal region from amino acids 39 to 196 on Zfp36 (Zfp36-mF) that can be stably expressed and specifically interacts with Pabpc1 (*Figure 3F*).

## Zfp36-mediated translational repression is dependent on Pabpc1

To test whether the Zfp36-Pabpc1 interaction is required for the Zfp36-mediated translational repression, we performed two experiments. First, using the λN polypeptide and the BoxB RNA motif tethering system, we tethered Zfp36 and 2 Zfp36 truncations that cannot interact with Pabpc1 (*Figure 4—figure supplement 1*), separately, to the 3'UTR of an *FLuc* mRNA, and then measured the luciferase activity, the *Fluc* mRNA level, and the translatability (*Figure 4A*). We found that unlike Zfp36, the two truncations did not specifically repress the translation of the *Fluc* mRNA (*Figure 4B*). This result is consistent with the notion that the Zfp36-Pabpc1 interaction is functional.

Next, we speculated that if the Zfp36-Pabpc1 interaction is important, then Zpf36 could not repress the translation of the transcripts that are devoid of Pabpc1, such as mRNAs without poly(A) tails. To test this hypothesis, we created a poly(A) minus (poly(A)-) mRNA by replacing the SV40 polyadenylation signal on the FLuc reporter (FLuc-5BoxB-SV40pA) with a sequence from the 3' end of MALAT1, which can result in cytoplasmic transcripts without poly(A) tails (*Figure 4C*) (*Wilusz et al., 2012*). Indeed, when the resulting *FLuc-5Box-MALAT1* mRNA was fractionated by oligod(T)

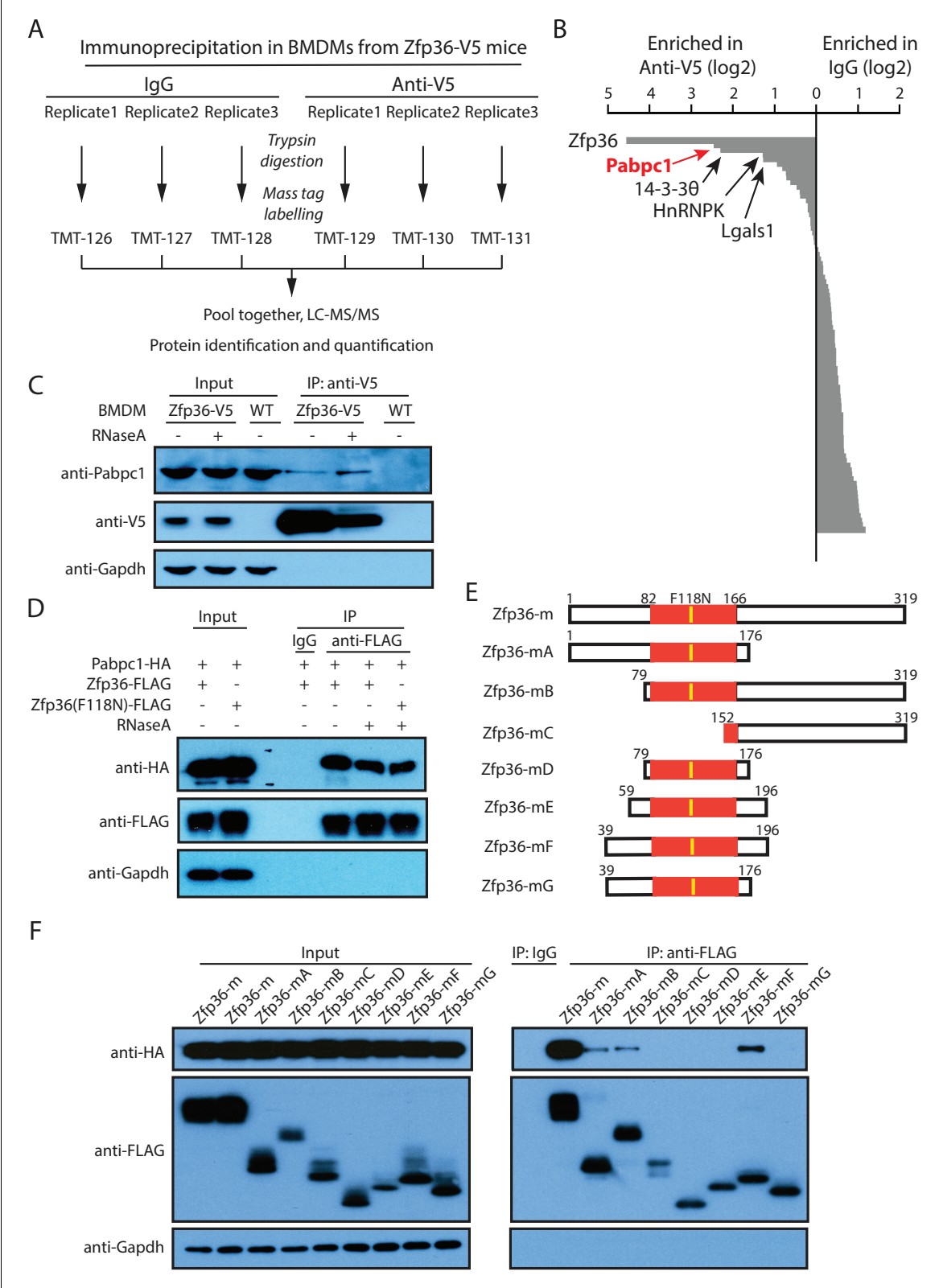

**Figure 3.** Endogenous Zfp36 interacts with Pabpc1 in LPS-stimulated BMDM. (**A**) Workflow for the tandem mass tag (TMT) quantitative proteomics to identify the proteins associated with endogenous Zfp36 in LPS-stimulated BMDMs. (**B**) Zfp36-associated proteins identified by the TMT quantitative proteomics. (**C**) Zfp36 interacts with endogenous Pabpc1 in LPS-stimulated BMDMs. BMDMs from the Zfp36-V5 mice or wild-type (WT) mice were treated with 100 ng/ml LPS for 4 hr, followed by IP with (+) or without (-) RNaseA (200 ng/ml) using an anti-V5 antibody. The input (5%) and IP products

*Figure 3 continued on next page*

Figure 3 continued

were subject to SDS-PAGE and Western blot analysis using the indicated antibodies. (D) The Zfp36-Pabpc1 interaction is independent of RNA. The plasmid expressing the FLAG-tagged wild-type Zfp36 or a mutant, Zfp36(F118N), was co-transfected with a plasmid expressing the HA-tagged Pabpc1 into 293 T cells, respectively. IP was performed with (+) or without (-) RNaseA (200 ng/ul) using either an anti-FLAG antibody or an IgG. SDS-PAGE and Western blot were used to analyze the indicated proteins in the input and IP samples. (E) Schematic presentation of Zfp36 truncations for mapping the region interacting with Pabpc1. (F) Identification of the region on Zfp36 that binds Pabpc1. The plasmids expressing FLAG-tagged Zfp36 truncations shown in (E) were co-transfected with a plasmid expressing the HA-tagged Pabpc1 into 293 T cells, respectively. IP was performed with RNaseA (200 ng/ml) using an anti-FLAG antibody or an IgG. SDS-PAGE and Western blot were used to analyze the indicated proteins in the input and IP samples. The following figure supplement is available for *Figure 3*.

The following figure supplement is available for figure 3:

**Figure supplement 1.** The Zfp36-Pabpc1 interaction is independent of RNA.

magnetic beads, most (>95%) remained in the poly(A)- fraction, which is similar to poly(A)- transcripts, such as *histone* mRNA (*H2ab*) and 18S rRNA but different from the *Gapdh* transcript, a poly (A)+ mRNA (*Figure 4C*). To further determine that the poly(A)- *FLuc-5Box-MALAT1* mRNA is not associated with Pabpc1, we performed RNA IPs on the 293 T cells expressing an HA-Pabpc1 with either the *FLuc-5BoxB-SV40pA* mRNA or the *FLuc-5BoxB-MALAT1* mRNA. We found that the HA-Pabpc1 can pull down significant amounts of the poly(A)+ transcripts, such as the *FLuc-5BoxB-SV40pA* mRNA and *Gapdh* mRNA, but not the poly(A)- transcripts, such as the *FLuc-5BoxB-MALAT1* mRNA and *histone* mRNA (*H2ab*) (*Figure 4D and E*). These results indicated that Pabpc1 does not bind the poly(A)- *FLuc-5BoxB-MALAT1* mRNA. When Zfp36 was tethered to the *FLuc-5BoxB-MALAT1* mRNA, we found that no change in luciferase activity, *FLuc* mRNA level, or translatability compared with those from tethering a control protein (GFP) (*Figure 4F*), which indicates that Zfp36 cannot repress the translation of the poly(A)- mRNA that is not associated with Pabpc1. Together, these results argue for the functional importance of the Zfp36-Pabpc1 interaction in the Zfp36-mediated translational repression.

## Identification of global mRNA targets of Zfp36 in the activated BMDMs by CLIP-seq

To determine the functional relevance of the Zfp36-Pabpc1 interaction on the expression of Zfp36 target mRNAs, we first identified the mRNAs that endogenous Zfp36 binds in activated BMDMs. Specifically, we performed CLIP-seq (*Darnell, 2010*) in Zfp36-V5 BMDMs. In this method, the activated BMDMs were first crosslinked by UV (254 nm), which only introduces covalent bonds between RNA and proteins that are in direct contact with each other (*Darnell, 2010*). The protein-free regions of RNAs were digested by RNase1 in the cell lysate. Then, the Zfp36 and its associated RNA fragments were IPed by the V5 antibody. The covalent link between Zfp36 and its associated RNA fragments allows stringent washes during the IP step, which substantially decreases nonspecific interactions. The RNA fragments were then isolated and cloned for high-throughput sequencing. A total of ~5 million uniquely mapped reads were obtained from two independent CLIP-seq experiments (*Figure 5A*). Importantly, the CLIP results from these two biological replicas were similar to each other but different from RNA-seq on total RNA, which indicates high reproducibility of the data (*Figure 5B*).

We identified 280 mRNAs with clear Zfp36-binding peaks present in both of the two independent CLIP-seq experiments (*Supplementary file 3*). These mRNAs include well-characterized Zfp36 targets, such as *TNF*, *IL-6*, and *Zfp36* itself. Most of the binding peaks are located within the 3'UTRs of the target mRNAs (*Figure 5C*). Critically, motif analysis revealed that the most significantly enriched sequence motif within the Zfp36-binding peaks is UAUUUAUU (*Figure 5D*). Although UV-C (such as UV254nm) induced crosslinking preferentially occurs at uridines (*Sugimoto et al., 2012*), the U-rich motif we identified (*Figure 5D*) is consistent with the Zfp36-binding motifs determined from in vitro RNA-binding studies on Zfp36 (reviewed in *Brooks and Blackshear, 2013*), strongly arguing that the UAUUUAUU sequence is also the in vivo binding motif of Zfp36 in the activated BMDMs. Gene ontology analysis revealed that the Zfp36 target mRNAs are involved in several important pathways in inflammatory responses, such as cytokine and chemokine signaling pathways (*Figure 5E*).

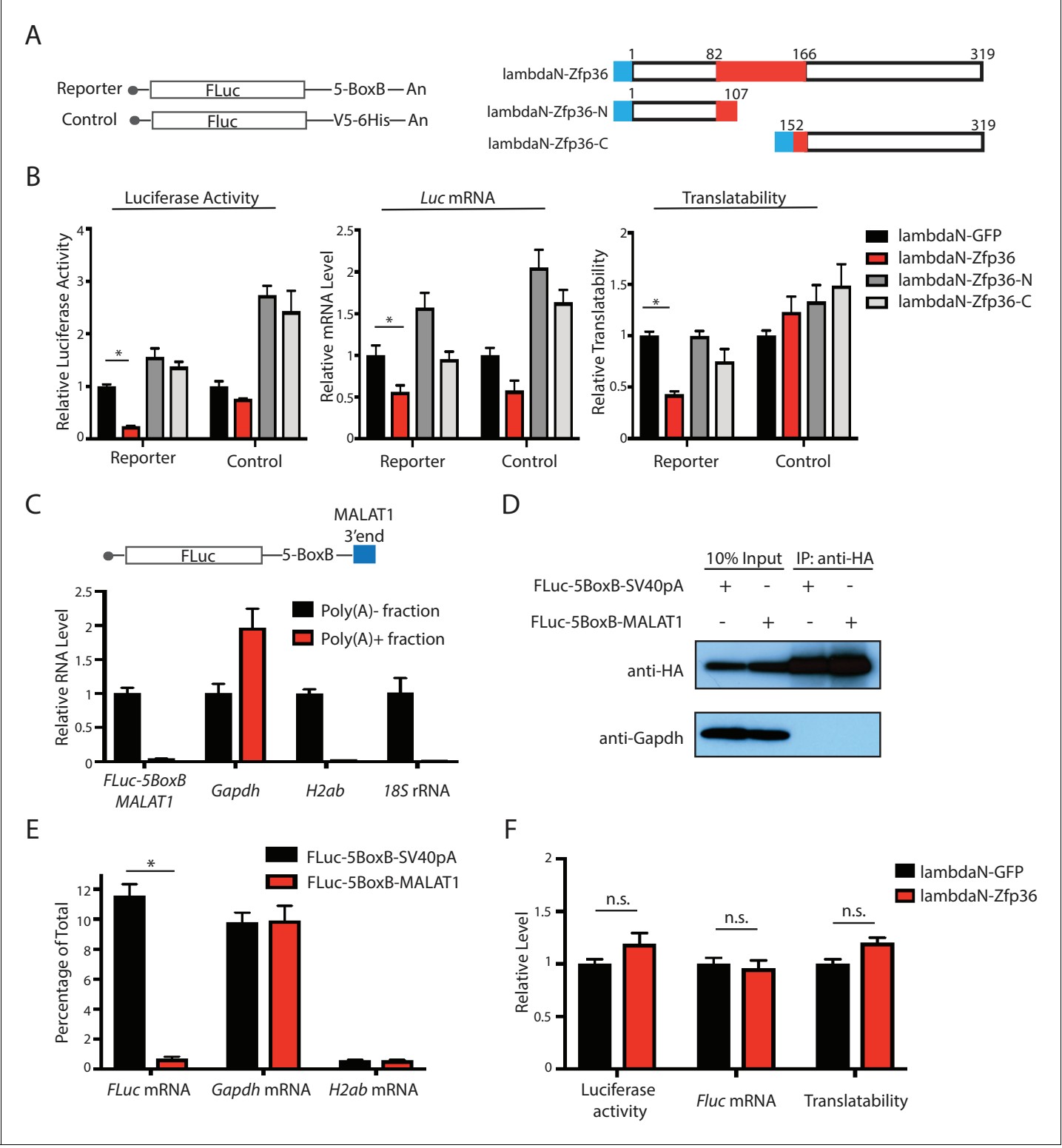

**Figure 4.** The Zfp36-Pabpc1 interaction is required for Zfp36-mediated translational repression. (A) Luciferase reporters and Zfp36 and its truncations used in the tethering experiment. The blue box represents the λN polypeptide. (B) Luciferase activity, the FLuc mRNA, and translatability determined in the tethering experiment. The FLuc-5BoxB reporter plasmid or the control plasmid was co-transfected with either the λN-GFP plasmid or plasmids expressing λN-Zfp36 and its truncations shown in (A) into 293 T cells, respectively. The luciferase assay and the mRNA measurement were performed at 24–30 hr post-transfection. The translatability was calculated as the luciferase activity normalized by the FLuc mRNA level. The luciferase activity, FLuc mRNA, and the translatability from the λN-GFP expressing cells were set as 1 for relative quantification, respectively. (C) The FLuc-5BoxB-MALAT1

*Figure 4 continued on next page*

eLIFE Research article                                                                                      Biochemistry

*Figure 4 continued*

reporter mRNA is a poly(A)- transcript. Total RNA from the 293 T cells transfected with the FLuc-5BoxB-MALAT1 reporter plasmid were fractionated by oligod(T)$_{25}$ magnetic beads. RNAs were quantified by qRT-PCR in the poly(A)+ and poly(A)-fractions. The RNA level in the poly(A)- fraction was set as 1 for relative RNA level calculation. (D) IP of HA-Pabpc1 in 293 T cells. A HA-Pabpc1 expressing plasmid was co-transfected into 293 T cells with either the FLuc-5BoxB-SV40pA reporter or the FLuc-5BoxB-MALAT1 reporter. IP was performed using an anti-HA antibody, and Western blot was used to examine the indicated proteins in the input and IP samples. (E) Pabpc1 does not bind the FLuc-5BoxB-MALAT1 reporter mRNA. qRT-PCR was performed on indicated mRNAs from the total RNA isolated from the input and IP samples of (D). (F) Zfp36 cannot repress the translation of the FLuc-5BoxB-MALAT1 mRNA. The luciferase, mRNA level, and translatability of the FLuc-5BoxB-MALAT1 mRNA were determined in the tethering experiment as described in (B). All results represent the means (± SD) of three independent experiments. *p<0.05, n.s. not significant (p>0.05) by the Student's t-test. The following figure supplement is available for *Figure 4*.

The following figure supplement is available for figure 4:

**Figure supplement 1.** The Zfp36 N-terminus and C-terminus fragments do not interact with Pabpc1.

Collectively, these results indicated the validity of the CLIP-seq data and identified mRNA targets that endogenous Zfp36 binds in the activated BMDMs.

## The Zfp36-Pabpc1 interaction is required for the Zfp36-mediated translational repression in activated BMDMs

To determine whether the Zfp36-Pabpc1 interaction is functionally relevant in controlling Zfp36 target mRNA expression in activated BMDMs, we first attenuated this endogenous interaction. Specifically, we expressed the minimal Zfp36 region (Zfp36-mF) that can interact with Pabpc1 (*Figure 3E,F*) in Zfp36-V5 BMDMs using a retroviral vector (*Figure 6A*). Importantly, the Zfp36-mF contains a point mutation (F118N) that abolishes the RNA-binding activity (*Lai et al., 2002*), which eliminates the complications of competitive binding for Zfp36 target mRNAs. IP with RNaseA indicated that the endogenous Zfp36-Pabpc1 interaction was decreased by half in the Zfp36-mF-expressing BMDMs compared with that in control BMDMs (*Figure 6B,C*).

Next, we monitored the ribosome association of Zfp36 target mRNAs by sucrose-density-gradient-mediated polysome profiling. Expression of the Zfp36-mF fragment did not change the global polysome profile in the activated BMDMs (*Figure 6D*), consistent with the notion that this fragment does not bind RNA. We then examined the distribution of seven Zfp36 target mRNAs across the gradient. These mRNAs are abundantly expressed and have strong Zfp36-binding peaks within their 3'UTRs. Upon weakening the Zfp36-Pabpc1 interaction, all seven of the Zfp36 target mRNAs had significant shifts from the mRNP region, which is devoid of translocating ribosome, to the polyribosome region, where active translation occurs, on the sucrose gradient (*Figure 6E*). Critically, the distribution of *Gapdh* mRNA, which is not a Zfp36 target, did not change upon Zfp36-mF expression (*Figure 6E*). These observations indicated that attenuation of the Zfp36-Pabpc1 interaction resulted in a specific increase of ribosome association on Zpf36 target mRNAs.

To further verify the functional importance of the Zfp36-Pabpc1 interaction, we measured the mRNA and protein levels of two Zfp36 targets: *TNF* and *IL-6*. These two cytokines are critical players in the inflammatory response. Quantitative RT-PCR revealed that *IL-6* mRNA was not significantly altered and *TNF* mRNA was slightly decreased (to ~70% of control) when the Zfp36-mF was expressed (*Figure 6F*). Interestingly, however, at the protein level, both TNF and IL-6 were significantly increased in the cell lysates from the Zfp36-mF expressing BMDMs (*Figure 6G*). Similar increases were observed in the supernatant from the Zfp36-mF-expressing cells (*Figure 6G*), although the increase in IL-6 was not statistically significant. Thus, the combined results from the mRNA level and the protein level indicated that the translation efficiency of these two Zfp36 target mRNAs increased when the endogenous Zfp36-Pabpc1 interaction was attenuated.

Collectively, these results revealed that the Zfp36-Pabpc1 interaction is required for the translational repression of Zfp36 target mRNAs in activated BMDMs.

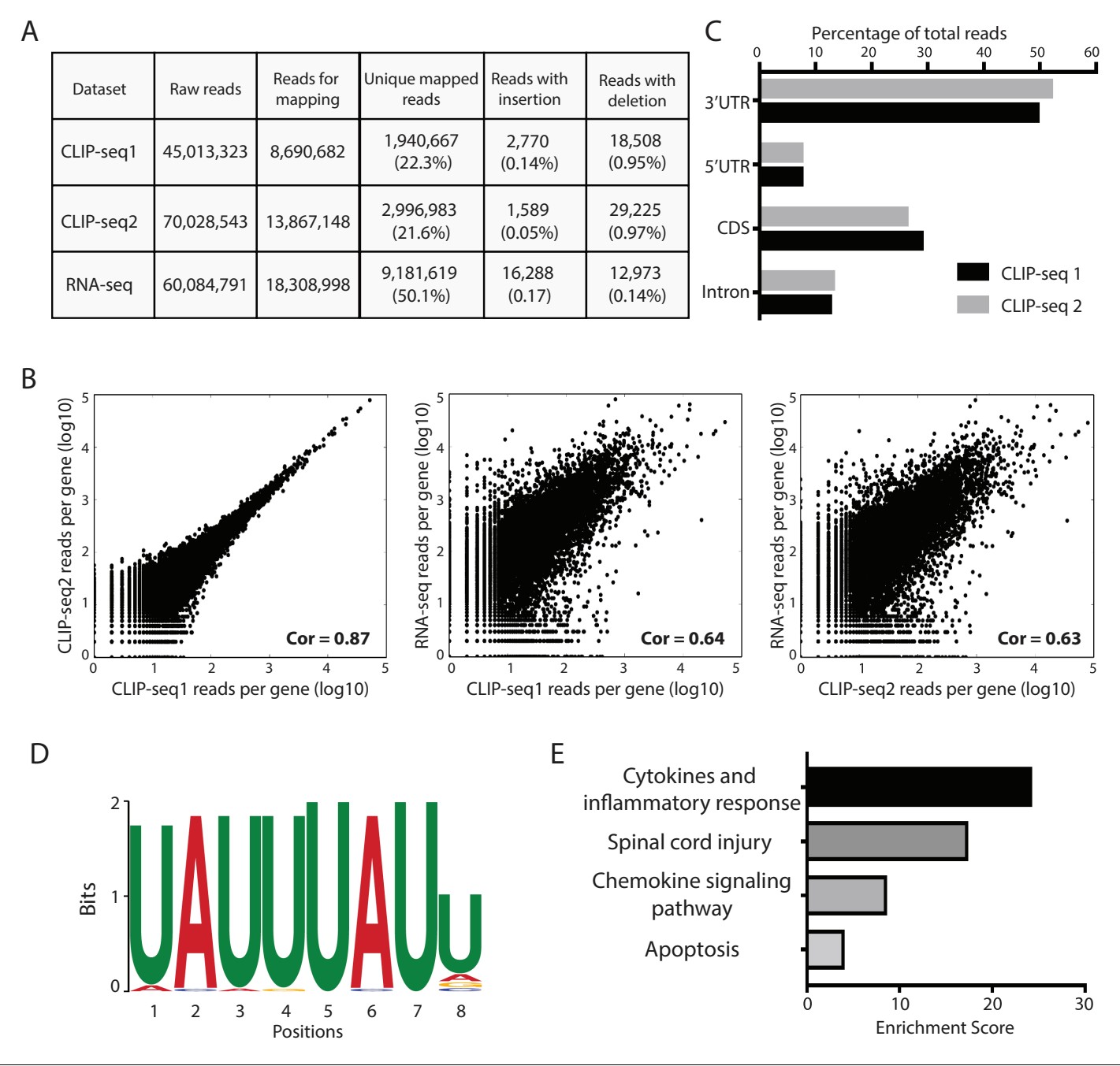

**Figure 5.** Transcriptome-wide identification of Zfp36 target mRNAs using CLIP-seq. (**A**) A Table summarizing the number of uniquely mapped reads to the mouse genome for the CLIP-seq duplicates and RNA-seq. (**B**) Log plots of the number of uniquely mapped reads per gene from the CLIP-seq duplicates and the mRNA-seq. Each dot represents a gene. Pearson's correlation coefficients (Cor) are indicated. (**C**) CLIP-seq reads distribution within protein coding genes. For the CLIP-seq reads uniquely mapped to mRNAs, their percentages in the 3'UTR, 5'UTR, CDS, and intron regions are shown. (**D**) Over-represented Zfp36-binding motifs identified by CLIP-seq. The motif was identified by MEME analysis on the Zfp36 binding clusters within mRNAs. The E-value is 1.6e-22. (**E**) Gene ontology analysis of the Zfp36 target mRNAs in activated BMDMs.

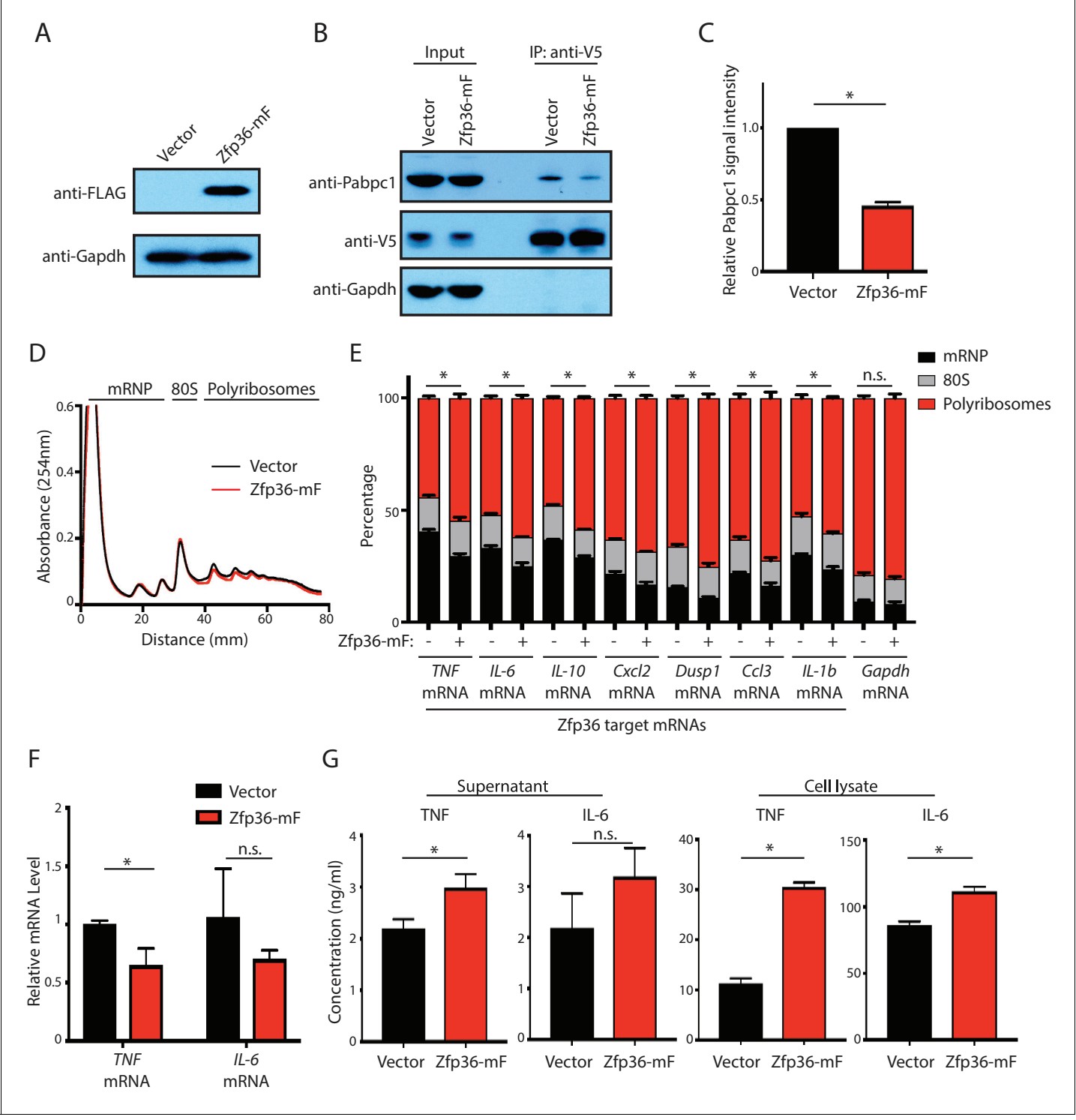

**Figure 6.** The Zfp36-Pabpc1 interaction is important for Zfp36-mediated translational repression in LPS-stimulated BMDM. (**A**) Expression of the Zfp36-mF fragment in LPS-stimulated BMDMs. BMDMs from the Zfp36-V5 mice were transduced with either an empty retroviral vector or a retroviral vector expressing the Zfp36-mF fragment (*Figure 3E*). The transduced BMDMs were treated with LPS for 4 hr, and the expression level of the Zfp36-mF was monitored by Western blot using the indicated antibodies. (**B**) Zfp36-mF attenuates the endogenous Zfp36-Pabpc1 interaction in activated BMDMs. The Zfp36 was IPed with RNaseA (200 ng/ml) from the cell lysates of the LPS-stimulated transduced BMDMs in (**A**) using an anti-V5 antibody. SDS-PAGE and Western blot were used to analyze the indicated proteins in the input and IP samples. (**C**) Quantification of the Zfp36-Pabpc1 interaction. The IPed endogenous Pabpc1 were quantified by the ImageJ, and the Pabpc1 intensity from the BMDMs expressing the empty vector was set as 1 for relative quantification. (**D**) Polysome analysis of Zfp36-mF expressing primary BMDMs. (**E**) Zfp36 target mRNAs are translationally up-regulated in Zfp36-mF

*Figure 6 continued on next page*

*Figure 6 continued*

expressing BMDMs. The mRNA distribution in the mRNP, the 80S, and the polyribosome fractions (shown in D) were quantified by qRT-PCR in the activated BMDMs expressing either Zfp36-mF (+) or an empty retroviral vector (-), respectively. (**F**) Zfp36 target mRNA levels were not dramatically changed in Zfp36-mF expressing BMDMs. TNF and IL-6 mRNAs were quantified by qRT-PCR in the activated BMDMs expressing either Zfp36-mF or an empty vector, respectively. 18S rRNA was used for normalization. (**G**) Proteins from Zfp36 target mRNAs are increased upon attenuating the Zfp36-Pabpc1 interaction. TNF and IL-6 proteins were quantified by ELISA in the cell lysate and supernatant of the Zfp36-mF expressing BMDMs and the control BMDMs, respectively. The results represent the means (± SD) of three independent experiments. *p<0.05, and n.s. not significant by the Student's t-test.

## Zfp36 represses translation at similar steps as Pabpc1 regulates translation

To determine which step(s) of translation Zfp36 interferes with, we used bicistronic reporters containing IRES (internal ribosome entry site) elements. IRES elements are structured RNAs present in many viruses that can position and activate eukaryotic translation initiation (*Fraser and Doudna, 2007*). Importantly, compared with canonical translation, different viral IRES elements require different subsets of translation initiation factors. Thus, by examining the sensitivity of various IRES-mediated translations to Zfp36, we aimed to dissect which step(s) of translation are repressed by Zfp36. Here, we used two IRES elements: the hepatitis C virus (HCV) IRES, which requires all the initiation factors except eIF4G and eIF4E for translation initiation, and the cricket paralysis virus (CrPV) IRES, which only needs the 40S ribosome subunit for translation initiation. These two IRES elements were inserted between an FLuc CDS and a renilla luciferase (RLuc) CDS on the bicistronic reporter, respectively (*Figure 7A*). Thus, FLuc translation is controlled by the canonical cap-dependent translation, whereas RLuc is translated by IRES-mediated translation. In addition, the BoxB RNA motifs in the 3'UTR of the reporter enable specific tethering of either Zfp36 or a control protein (GFP) on the bicistronic reporter mRNAs using the λN polypeptide. When Zfp36 was tethered, the reporter mRNA levels did not change compared with those of GFP tethering (*Figure 7B*). Interestingly, however, both the FLuc and the RLuc luciferase activities were significantly decreased in either the HCV-IRES reporter or the CrPV-IRES reporter (*Figure 7C*). Since the 40S ribosome subunit is the only factor needed by the CrPV IRES element to initiate translation (*Fraser and Doudna, 2007*), these results strongly argue that Zfp36 either interferes with 40S ribosome subunit recruitment during translation initiation or inhibits the events at or after the 60S subunit joining step during mRNA translation. Interestingly, Pabpc1 also regulates these steps in facilitating mRNA translation (*Mangus et al., 2003*). Thus, these observations indicate that Zfp36 represses translation at similar step(s) as Pabpc1 regulates translation.

## Discussion

Gene expression is precisely regulated during the inflammatory responses for controlling infections and limiting detrimental effects of inflammation. Previous studies identified two two layers of regulation at the transcriptomic level. First, in the early stage of inflammation, several critical transcriptional factors are activated to rapidly produce mRNAs of inflammatory response genes (reviewed in *Medzhitov and Horng, 2009*). Second, in the late stage of inflammation, mRNA stability regulatory circuits promote the degradation of mRNAs of many inflammatory cytokines thus preventing overproduction of cytokines that are potentially toxic to normal tissues and thereby contributing to resolving inflammation (reviewed in *Carpenter et al., 2014*). Using parallel ribosome profiling and RNA-seq, here we identified hundreds (724) of differentially translated mRNAs in the primary BMDM-mediated inflammatory response. We think that this number may be an underestimate of mRNAs regulated at the translation level due to rapid degradation of translationally repressed mRNAs. In contrast to a few unique biological systems, such as oocytes, in which mRNA decay activity is minimal, in most somatic cells it is challenging to fully identify all the translationally regulated mRNAs at a steady state due to efficient and dynamic mRNA degradation. Nonetheless, among the 3'UTRs of the differentially translated mRNAs we identified, there are significantly enriched binding motifs of several RNA-binding proteins expressed in the activated BMDMs. These matched *cis-*

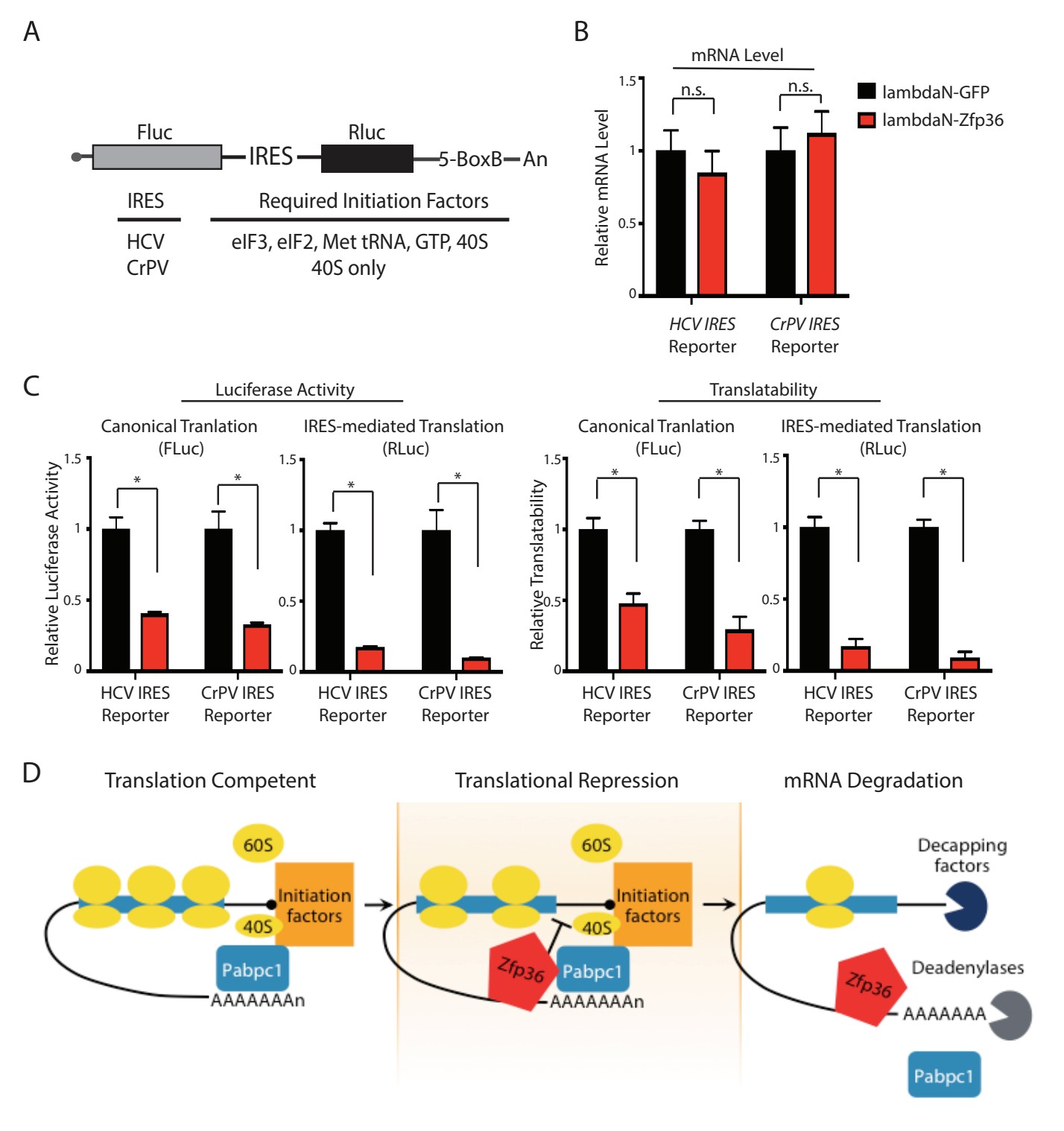

**Figure 7.** Zfp36 represses translation at similar steps as Pabpc1 regulates translation. (**A**) Schematic presentation of the bicistronic luciferase reporter system for dissecting how Zfp36 represses translation. (**B**) Tethering Zfp36 does not change the mRNA levels of the bicistronic reporters. The bicistronic reporter plasmid (either the HCV reporter or the CrPV reporter) was co-transfected with the λN-GFP plasmid or the λN-Zfp36 plasmid into 293 T cells, respectively. The reporter mRNA levels from the transfected 293 T cells were quantified by qRT-PCR, and 18S rRNA was used for normalization. (**C**) Zfp36 inhibits both canonical translation and IRES-mediated translation. The bicistronic reporter plasmid was co-transfected with either the λN-GFP plasmid or the λN-Zfp36 plasmid into 293 T cells, respectively. The firefly (FLuc) and renilla (RLuc) luciferase activities were measured at 24–30 hr post-transfection. The translatability is calculated as the luciferase activity normalized by the reporter mRNA level. (**D**) A model for Zfp36-mediated regulation

*Figure 7 continued on next page*

*Figure 7 continued*

of gene expression in activated BMDMs. All the quantification results represent the means (± SD) of three independent experiments. *p<0.05, and n.s. not significantly different by the Student's t-test.

elements and *trans*-acting factors strongly argue for the presence of translational regulatory networks during the inflammatory response.

Zfp36 is an important factor we characterized in these translational regulatory networks. Previous studies from multiple labs indicate that Zpf36 not only is an inducer of mRNA degradation by recruiting mRNA decay factors, such as mRNA deadenylases and decapping factors (reviewed in *Brooks and Blackshear, 2013*), but also can function as a translational silencer (*Brooks and Blackshear, 2013*; *Franks and Lykke-Andersen, 2007*; *Fu et al., 2016*; *Qi et al, 2012*; *Tao and Gao, 2015*; *Tiedje et al., 2016*). Based on the results predominantly from cell lines and overexpression systems, multiple mechanisms of Zfp36-mediated translational repression have been proposed. For example, Zfp36 can repress target mRNA translation via interacting with the translation initiation factor 4E2 (*Tao and Gao, 2015*), binding the mRNA decapping activator RCK/p54 (*Qi et al, 2012*), recruiting the 4EHP-GYF2 cap-binding complex (*Fu et al., 2016*), and delivering target mRNAs to processing bodies (*Franks and Lykke-Andersen, 2007*). Here, we focus on how endogenous Zfp36 regulates gene expression in mouse primary BMDMs. Using a knock-in mouse with a V5 epitope tag sequence inserted in-frame near the stop codon of Zfp36, we found that Pabcp1, the cytoplasmic poly(A) tail binding protein, is the most abundant RNA-independent binding partner with the Zfp36 in primary activated BMDMs. Critically, the Zfp36 expressed in the knock-in mouse is at its endogenous level, because the 51-nt knock-in V5 tag sequence neither changes the Zfp36 promoter nor eliminates the regulatory elements in the 3'UTR. Thus, this Zfp36-Pabpc1 interaction is an endogenous protein interaction in activated BMDMs. Surprisingly, however, we did not detect any mRNA decay factors or the previously identified proteins involved in translational repression in our quantitative proteomic analysis on endogenous Zfp36. How can this result be reconciled with previous findings on the proteins associated with Zfp36?

We think that this difference can be explained by the several possibilities. First, previous observations of the interactions between Zfp36 and mRNA decay factors and translational regulators were predominantly from the experiments of Zfp36 overexpression in cell lines, followed by either Co-IP or IP and mass spectrometry; or yeast two-hybrid screens (reviewed in *Brooks and Blackshear, 2013*). These results indicate that Zfp36 can interact with those factors. Our study, however, we focused primarily on the protein(s) that interact with endogenous Zfp36 in primary activated BMDMs. The differences in both the biological setting and the Zfp36 expression levels may contribute to the different results. Second, it is also possible that the interaction between Zfp36 and mRNA decay factors is transient and beyond detection by the quantitative proteomics we used. Consistent with this notion, mRNA decay is a highly dynamic process. Once the mRNA is destroyed, the decay factors will be released and assemble on the next transcript. Thus, it is likely that Zpf36 binds to the mRNA decay factors within a very specific time window during the life of the target mRNAs, whereas the proteins we identified are steady-state partners that interact with endogenous Zfp36 in primary activated BMDMs.

The Pabpc1-interacting region on Zfp36 overlaps with its RNA-binding domain. Critically, we confirmed that the Zfp36-Pabpc1 interaction is not mediated by RNA. Interestingly, the Pabpc1-interacting domain (in the middle) and the domains that can interact with mRNA decay factors (at the N-terminus and the C-terminus) are located in different regions on Zfp36. This observation suggests that Zfp36's binding to Pabcp1 to repress mRNA translation (which is characterized in this study) and recruiting decay factors to promote mRNA degradation (which is described in the literature (reviewed in *Brooks and Blackshear, 2013*) are two different steps. Considering that mRNA translation repression can accelerate mRNA degradation (reviewed in *Coller and Parker, 2004*; *Richter and Coller, 2015*), we speculate that these two steps are intimately related and may occur in a sequential order in the Zfp36-mediatd regulation of gene expression. Thus, we propose the following model (*Figure 7D*): once bound to target mRNAs, Zfp36 interacts with Pabcp1 to attenuate translation; then mRNA decay factors are recruited to destroy the transcripts. Thereby, Zfp36 facilitates the transition of its target mRNAs from a translation-competent state in the initial stage of

inflammation to a degradation-prone state in the late stages of the inflammatory response. Since many Zfp36 target mRNAs are proinflammatory cytokines, such as *TNF*, we speculate that this Zfp36-mediated transition of its target mRNAs from an active state to an inactive state of protein production facilitates resolving inflammation.

Targeting poly(A)-binding proteins by sequence-specific RBPs to repress mRNA translation is a new mechanism of translational control. Unlike our knowledge of regulating mRNA translation through the classical translational initiation factors, such as eIF4E and eIF4G, our understanding of modulating translation via Pabpc1 is very limited. Here, we determined that the Zfp36-Pabpc1 interaction is required for the Zfp36-mediated translational repression in primary activated BMDMs. Using reporter assays, we found that Zfp36 represses translation at either late stage(s) of initiation or postinitiation step(s). This result is consistent with Pabpc1's roles in translation by enhancing ribosome recruitment to the mRNA at both the 40S ribosome subunit recruitment step and the 60S subunit joining step (*Kahvejian et al., 2005*; *Mangus et al., 2003*). Thus, we speculate that Zfp36 represses mRNA translation by compromising Pabpc1's ability to maintain mRNA in a translation-competent state. Interestingly, Pabpc1 functions at the interface of controlling mRNA translation and stability by protecting mRNA from deadenylases and stimulating mRNA translation (*Mangus et al., 2003*). These unique functions make Pabpc1 an ideal target for shutting down gene expression posttranscriptionally. We think that in the activated BMDMs, through modulating Pabpc1, Zfp36 can rapidly inhibit the translation of its target mRNAs and promote their degradation to resolve inflammation.

Among the 280 Zfp36 target mRNAs identified by CLIP-seq in the activated BMDMs, we noticed that not every one is translationally repressed during the inflammatory response (the differentially translated genes in the *Supplemental file 1* versus the mRNAs listed in the *Supplemental file 3*). This can be explained by several possibilities. First, it is possible that multiple RBPs can bind on the same mRNA, and the ultimate fate of the transcript is determined by the combinatory functions of all the associated proteins. Indeed, besides Zfp36, we also predicted several other RBPs as potential translation regulators. Thus, it would be interesting to explore the functional interactions between Zfp36 and other RBPs that can regulate the same transcripts. These results will provide insight into how Zfp36's activity is antagonized or stimulated during the inflammatory response. Second, not every Zfp36's binding event may result in expression changes of the target mRNAs. Third, when we determined the differentially translated mRNAs during the inflammatory response, we focused on the mRNAs with the expression levels $\geq$ 10 FPKM at all the time points. We set this threshold because the apparent TEs (Ribo-seq divided by RNA-seq) of lowly expressed mRNAs have large variations between the two biological replicate samples. Although have high expression levels at late time-points in the inflammatory response, some of the Zfp36 target mRNAs, such as the *IL-6* mRNA and the *IL-1b* mRNAs, are expressed at low level (<10 FPKM) at 0 hr. Thus, these transcripts are excluded from the list of the identified differentially translated mRNAs.

It is important to mention that the mechanistic insights we obtained regarding Zfp36-mediated regulation of gene expression are different from previous studies in Zfp36 overexpressed cell lines (*Franks and Lykke-Andersen, 2007*; *Fu et al., 2016*; *Qi et al, 2012*; *Tao and Gao, 2015*). The Zfp36-V5 knock-in mouse allowed us to focus on the endogenous Zfp36 in primary innate immune cells, making the molecular mechanisms from this study biologically relevant. We believe similar approaches can be applied to characterize molecular mechanisms of other RBPs that lack high-quality antibodies under biologically relevant settings.

In summary, our study revealed widespread translational regulations of the primary macrophage-mediated inflammatory response. We found that Zfp36, a critical RBP for resolving inflammation, mechanistically functions as a translation repressor by targeting the poly(A)-tail binding protein, Pabpc1, to inhibit proinflammation gene production. Besides Zfp36, we also identified additional RBPs that may be involved in translational regulatory networks during the inflammatory response. Future functional and mechanistic characterization of these RBPs will provide novel insights into gene expression in the innate immune response.

## Material and methods

The plasmids, oligos, antibodies, chemicals used in this study are listed in *Supplementary file 4*.

## Zfp36-V5 knock-in mouse

The Zfp36-V5 knock-in mouse was generated via CRISP/Cas9-mediated genomic editing using the protocols described previously (*Wang et al., 2013*). Specifically, Cas9 mRNA (TriLink BioTechnologies, San Diego, CA) (100 ng/µl), sgRNA derived from oWH2056 (50 ng/µl), and a donor oligo (owH2057) containing the V5 tag sequence surrounded by a 60nt homologous arm on each side (50 ng/µl), were co-injected into mouse zygotes. Then the injected zygotes were transferred to mouse foster mothers. The resulting mice were genotyped at 3–4 weeks old to check the V5 tag integration at the Zfp36 locus.

## Cell lines, BMDMs culture and retrovirus transduction

In this study, we predominantly used primary cells (that is, mouse BMDMs) isolated from 8 to 12 weeks B6 mice except for the 293 T cells used in the transfection-based assays. The 293 T cells used in this study were obtained from ATCC (ATCC CRL-11268) (RRID:CVCL_0063). The cell identity was authenticated via short tandem repeat (STR) profiling. Mycoplasma contamination was tested as negative by a PCR method.

BMDMs isolation, culture, and retroviral transduction were performed in accordance with the published protocols (*Weischenfeldt and Porse, 2008*).

All the procedures on the isolation of BMDMs from mice were performed under a protocol (A35015) approved by the Mayo Clinic animal welfare committee.

## RNA isolation and quantitative RT-PCR

The RiboZol RNA Extraction Reagent (Amresco, Solon, OH) was used for total RNA isolation. The residual genomic DNA was removed by DNase1 (NEB, Ipswich, MA) digestion. Reverse transcription was performed using the SuperscriptII (Invitrogen, W a l t h a m , M A ) and random hexamers. Quantitative PCR was performed using the 2 X SYBR Green qPCR Master Mix (BioRad, Hercules, CA) on a Bio-Rad CFX Real-Time PCR Detection System.

## Immunoprecipitation and mass spectrometry

The Protein G Dynabeads (LifeTechnologies, Carlsbad, CA) were used for all the immunoprecipitation assays in this study. The TMT-based quantitative mass spectrometry was performed at the proteomic core facility of the Whitehead Institute for Biomedical Research (Cambridge, MA, USA).

## CLIP-seq and ribosome profiling

CLIP-seq was performed using the previously described protocols (*Moore et al., 2014*) with the following modifications: First, the BMDMs were crosslinked with 400 mJ/cm$^2$ UV254. Second, the IPed protein complexes were washed using the high stringent conditions.

Ribosome profiling was performed as previously described (*Ingolia et al., 2012*) except that we used a sucrose density gradient (5–50% w/v) to isolate the ribosome and the ribosome protected fragments. The computational analysis was performed in accordance with the pipelines described previously (*Alvarez-Dominguez et al., 2017*).

The sequencing data from these two sets of experiments are deposited at GEO (Gene Expression Omnibus) with the accession number GSE99787.

## Polysome analysis

Polysome analysis was performed as described previously (*Hu et al., 2014*). Briefly, BMDM cells were lysed in polysome lysis buffer (10 mM Tris-HCl, pH 7.4, 12 mM MgCl$_2$, 100 mM KCl, 1% Tween-20, and 100 mg/ml cycloheximide), and then the cell lysate was clarified by centrifugation at 21,130 g for 10 min at 4°C. The resulting supernatant was loaded onto a 5–50% (w/v) linear sucrose-density gradient and centrifuged at 39,000 rpm (260,000 g) for 2 hr at 4°C in a Beckman SW-41Ti rotor. The gradient was fractionated using a Gradient Station (BioComp) with an ultraviolet detector (Bio-Rad EM-1). RNA from each collected fraction was isolated using the method described above.

## Computational analysis

### Ribo-seq and RNA-seq analysis

The Ribo-seq and RNA-seq reads were trimmed by using fastq_quality_trimmer of the FASTX-Toolkit (http://hannonlab.cshl.edu/fastx_toolkit/index.html). The resulting reads were aligned to reference rRNA, tRNA and mitochondrial DNA sequences by using Bowtie v.1.1.1 (*Langmead et al., 2009*) with '–un' parameter to keep the unmapped reads. The per-processed RNA-seq reads were mapped by TopHat v2.1.1 to the mm10 reference genome with Refseq annotations by using parameters '–min-anchor 5 –segment-length 20 –read-mismatches 1 –no-novel-juncs'. The reads count of RNA-seq and Ribo-seq were calculated by using HTseq with parameters '-a 10 m union'. The raw reads counts were normalized as Reads Per Kilobase of transcript per Million mapped reads (RPKM). Only genes with RPKM greater than 10 at all the stages in RNA-seq and Ribo-seq were retained. The reproducibility between the RFP of replicates was visualized and checked by R package 'corrplot' and 'pysch'.

### Codon periodicity analysis

Codon periodicity of Ribo-seq and RNA-seq reads was assessed for known mRNAs in Refseq mm10 annotations. For annotations, only AUG-initiated CDSs were considered. Only the mapping sites of uniquely and perfectly aligned 28nt RFPs were selected to form bed files. The overlap between the mapping sites and CDS annotation was calculated by using the intersectBed function of bedtools.

### Translation efficiency analysis

Gene-level translation efficiency between each pair of adjacent stages was quantified by using the Babel analytical method (*Olshen et al., 2013*). Genes with empirical p-values<0.05 at any of the stages were considered to have differentially regulated translation efficiency. The translation efficiency at each stage was calculated as the ratio of normalized RPF density (RPKM) to the normalized RNA-seq reads density. The differentially translated genes were grouped by the 'Direction' in Babel output at the stage with minimum p-value. The translation efficiency of grouped genes was visualized as heat map using 'heatmap' R package with 'scale=row' parameter.

### RNA-binding protein motif enrichment analysis

The inferred RNA-binding proteins motifs of Mus musculus from CISBP-RNA Database (http://cisbp-rna.ccbr.utoronto.ca/) were used as the motifs database in the searching. The 3'UTR sequence of the longest CDS of mRNAs with differentially regulated TE was used as the sequence database. FIMO tool in the MEME Suite was used to search the occurrences of the motifs in the sequence database with the parameter '–norc' to ignore the reverse complement DNA strand. Zero-order background files for the motif searching were generated by fasta-get-markov tool in the MEME Suite. Ten shuffled copies were generated by using the fasta-shuffle-letters function in MEME Suite preserving the base composition and sequence length, then scanned by using FIMO with the same parameters. For each motif separately, Fisher's exact test was performed to estimate the odds ratio and the empirical p-value of enrichment. The motifs with false discovery rate adjusted p-values<0.05 were considered as significantly enriched.

### CLIP-seq analysis

The CLIP-seq reads were quality-checked by using FastQC (http://www.bioinformatics.babraham.ac.uk/projects/fastqc/). The reads mapped to rRNA, tRNA and mitochomdrial DNA sequences using Bowtie were removed from datasets. The resulting reads were mapped to the mm10 genome by using STAR with the parameters suggested in (*Van Nostrand et al., 2016*). Piranha (*Uren et al., 2012*) was used to call peaks from the two replicates over the input control datasets with parameters '-b 50 s -p 0.01'. The peaks were merged using mergePeaks function in HOMER if the distance between peak centers is less than or equal to 100nt. The merged peaks were then annotated by annotatePeaks tool in HOMER to the mm10 RefSeq mRNAs. The RNA sequences of the resulting peaks were obtained using getfasta (http://bedtools.readthedocs.io/en/latest/content/tools/get-fasta.html) in bedtools with parameter '-s'. MEME was used to search for motifs in the sequences of Clip-seq peaks with parameter '-w 8'. The GO term analysis of the mRNAs with Clip-seq peaks was performed by Enrichr (http://amp.pharm.mssm.edu/Enrichr/).

## Acknowledgements

We thank Dr. Juan R Alvarez-Dominguez for critical comments and Dr. Alyssa Quiggle for English editing on this manuscript. This work is supported by a start-up fund from Mayo Foundation for Medical Education and Research; a pilot grant from the Center for Biomedical Discovery at Mayo Clinic, a grant from the NHLBI (R00HL118157) (WH), and a grant from the NIGMS (R01GM102515) (SZ).

## Additional information

### Funding

| Funder | Grant reference number | Author |
| --- | --- | --- |
| Mayo Foundation for Medical Education and Research | | Wenqian Hu |
| National Heart, Lung, and Blood Institute | R00HL118157 | Wenqian Hu |
| National Institute of General Medical Sciences | R01GM102515 | Shaojie Zhang |

The funders had no role in study design, data collection and interpretation, or the decision to submit the work for publication.

### Author contributions

XZ, Data curation, Formal analysis, Investigation, Methodology, Writing—review and editing; XC, Data curation, Formal analysis, Methodology, Writing—review and editing; QL, Data curation, Formal analysis, Investigation, Writing—review and editing; SZ, Data curation, Formal analysis, Supervision, Writing—review and editing; WH, Conceptualization, Data curation, Formal analysis, Supervision, Funding acquisition, Investigation, Methodology, Writing—original draft, Project administration

### Author ORCIDs

Shaojie Zhang, http://orcid.org/0000-0002-4051-5549
Wenqian Hu, http://orcid.org/0000-0003-3577-3604

### Ethics

Animal experimentation: All the procedures on the isolation of BMDMs from mice were performed under a protocol (A35015) approved by the Mayo Clinic animal welfare committee.

## Additional files

### Supplementary files

• Supplementary file 1. TEs of mRNAs during the BMDM-mediated inflammatory response. The first tab listed the TEs of the differentially translated mRNAs during the inflammatory response. The second tab listed the TEs of all the mRNAs during the inflammatory response.

• Supplementary file 2. Proteins identified by the TMT proteomic analysis.

• Supplementary file 3. Target mRNAs of the endogenous Zfp36 in activated BMDMs.

• Supplementary file 4. Plasmids, antibodies, chemicals, and oligoes used in this study.

### Major datasets

The following dataset was generated:

**Database, license,**

| Author(s) | Year | Dataset title | Dataset URL | and accessibility information |
|---|---|---|---|---|
| Xu Zhang, Xiaoli Chen, Qiuying Liu, Shaojie Zhang, Wenqian Hu | 2017 | Global mRNA translational landscapes in mouse bone-marrow-derived-macrophage-mediated response to LPS (lipopolysaccharide) | https://www.ncbi.nlm.nih.gov/geo/query/acc.cgi?acc=GSE99787 | Publicly available at the NCBI Gene Expression Omnibus (accession no: GSE99787) |

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
