## [Decision Letter]

Thank you for submitting your article "Translation repression via modulating the cytoplasmic poly(A) binding protein in inflammatory response" for consideration by *eLife*. Your article has been favorably evaluated by James Manley (Senior Editor) and three reviewers, one of whom is a member of our Board of Reviewing Editors. The following individual involved in review of your submission has agreed to reveal his identity: Georg Stoecklin (Reviewer #3).

The reviewers have discussed the reviews with one another and the Reviewing Editor has drafted this decision to help you prepare a revised submission.

Zhang et al. analyze changes in mRNA translation during macrophage activation, and examine the mechanism by which the AU-rich element (ARE)-binding protein Zfp36 suppresses translation. Through an elegant knock-in strategy, they generate mice expressing V5-tagged Zfp36 from the endogenous locus. This allows them to identify under physiological conditions in primary bone-marrow derived macrophages both RNA targets of endogenous Zfp36 by CLIP, and protein interaction partners of endogenous Zfp36 by IP/MS. While the CLIP results confirm the known binding site and targets of Zfp36, the mass spec results suggest that Zfp36 interacts through a protein-protein interaction with Pabpc1, the major cytoplasmic poly-A RNA binding protein. The authors further narrow down the interacting region as the double zinc finger domain of Zfp36 with some flanking sequence on both sides. Based on a deletion mutant of Zpf36 that does not interact with Pabpc1, they propose that Zfp36 suppresses the translation of target mRNAs via its interaction with Pabpc1. This results it corroborated by overexpression of a Zfp36 fragment that reduces the interaction of endogenous full-length Zfp36 with Pabpc1.

The study shows potential for progressing the field of cytoplasmic regulation of gene expression. The mouse knock-in approach has pioneering character and should inspire other researchers in the field. Yet, there are some unresolved mechanistic issues that would need to be addressed before publication in *eLife*. Also, there appears to be a substantial lack of scholarship in terms of citing earlier work.

The principal evidence for the proposed mechanism is a Zfp36 mutant (mF) that contains large deletions of the N- and C-termini. This mutant is not specific for Pabpc1, and based on published reports will also be deficient for many other interactions. Deletion of the N-terminus will abolish binding of GYF2/4EHP, which was reported to mediate translational silencing of TTP targets. Deletion of the C-terminus will abolish binding to the Ccr4-Not complex, which may also mediate translational silencing given its role in the translational repression of miRNA-bound mRNAs. Hence, the Zfp36-mF mutant is by no means specific, and does not prove that binding of Pabpc1 is critical for mediating translational suppression. Given the known role of Pabpc1 as a general enhancer of translation, it is difficult to conceive how it would act as a translational suppressor when bound to Zpf36. A key experiment the authors could carry out is an in vitro translation assay, where recombinant Zfp36 would specifically suppress translation of an ARE-containing mRNA only upon addition of recombinant Pabpc1, or fail to do so upon depletion of endogenous Pabpc1. Another experiment that would provide some additional support for the proposed mechanism (though not as conclusive as the in vitro translation assay) is if tethering of the Zfp36-mF deletion mutant would be sufficient to cause translational suppression. This mutant most likely does not interact with GYF2/4EHP or the Ccr4-Not complex, although it may still bind other proteins in addition to Pabpc1.

The role of Zfp36 as a translational silencer (in addition to its role as an inducer of mRNA degradation) has been reported previously, through the work of the Gaestel, Brooks and Lykke-Andersen labs. The authors fail to mention any of these studies, which is, at best, careless. The authors need to quote these publications adequately and discuss their findings in the light of these previous studies.

---

## [Author Response]

*[…] The study shows potential for progressing the field of cytoplasmic regulation of gene expression. The mouse knock-in approach has pioneering character and should inspire other researchers in the field. Yet, there are some unresolved mechanistic issues that would need to be addressed before publication in eLife. Also, there appears to be a substantial lack of scholarship in terms of citing earlier work.*

*The principal evidence for the proposed mechanism is a Zfp36 mutant (mF) that contains large deletions of the N- and C-termini. This mutant is not specific for Pabpc1, and based on published reports will also be deficient for many other interactions. Deletion of the N-terminus will abolish binding of GYF2/4EHP, which was reported to mediate translational silencing of TTP targets. Deletion of the C-terminus will abolish binding to the Ccr4-Not complex, which may also mediate translational silencing given its role in the translational repression of miRNA-bound mRNAs. Hence, the Zfp36-mF mutant is by no means specific, and does not prove that binding of Pabpc1 is critical for mediating translational suppression. Given the known role of Pabpc1 as a general enhancer of translation, it is difficult to conceive how it would act as a translational suppressor when bound to Zpf36. A key experiment the authors could carry out is an in vitro translation assay, where recombinant Zfp36 would specifically suppress translation of an ARE-containing mRNA only upon addition of recombinant Pabpc1, or fail to do so upon depletion of endogenous Pabpc1. Another experiment that would provide some additional support for the proposed mechanism (though not as conclusive as the in vitro translation assay) is if tethering of the Zfp36-mF deletion mutant would be sufficient to cause translational suppression. This mutant most likely does not interact with GYF2/4EHP or the Ccr4-Not complex, although it may still bind other proteins in addition to Pabpc1.*

The fundamental question the reviewers asked here is whether or not Zfp36-mediated translational repression is dependent on Pabpc1.

Pabpc1 is the cytoplasmic poly(A) tail binding protein that associates with most of mRNAs. Importantly, perturbation on Pabpc1 may have pleiotropic effects to the cell. In the manuscript, we addressed the question via an system through modulating the poly(A) tail of a reporter mRNA. Specifically, we first generated an mRNA *without* the poly(A) tail by replacing the SV40 polyadenylation signal (SV40pA) on the FLuc reporter with a sequence from the 3’ end of MALAT1 (Figure 4). Previous studies by Wilusz et al., 2012 showed that fusing the 3’ end sequence of MALAT1 can result in a poly(A) minus transcript stably present in the cytoplasm in mammalian cells. Indeed, when we fractionated the mRNAs via an oligod(T) column, we found that the Fluc mRNA with the 3’end MALAT1 sequence is predominantly in the poly(A) minus fraction, which is similar to the histone2ab mRNA, a known poly(A) minus transcript, but different from the Gapdh mRNA, a poly(A)+ transcript (Figure 4). This result indicated that the Fluc-5BoxB-MALAT1 transcript we generated is a truly poly(A) minus mRNA. Next, we performed the tethering assay on this poly(A) minus transcript. We found that when tethered, Zfp36 fails to inhibit the translation of this poly(A) minus transcript (Figure 4). This result is different from tethering Zfp36 to a poly(A) plus transcript (Figure 4). Collectively, these observations indicate that Zfp36-mediated translational repression is dependent on the presence of the poly(A) tail.

We realize that although seems obvious, the poly(A) minus Fluc-5BoxB-MALAT1 transcript is devoid of Pabpc1 was still an assumption. To directly determine that Pabpc1 is not associated with the poly(A) minus Fluc-5BoxB-MALAT1 transcript we generated, we performed additional experiments. Specifically, we introduced a HA-Pabpc1 expressing plasmid with either the FLuc-5BoxB-SV40pA reporter or the Fluc-5BoxB-MALAT1 reporter into 293T cells. Then we performed IP on the HA-Pabpc1 from the transfected 293T cells (Figure 4) followed by qRT-PCR to determine whether the poly(A) minus Fluc-5BoxB-MALAT1 transcript is associated with Pabcp1 (Figure 4). We found that the Pabpc1 binds the poly(A)+ transcripts, such as the Gaphd1 mRNA and the FLuc-5BoxB-SV40pA transcript, but not the poly(A)- transcripts, such as the Histone2AB mRNA and the Fluc-5BoxB-MALAT1 transcript (Figure 4). This experiment indicated that the FLuc-5BoxB-MALAT1 transcript is indeed devoid of Pabpc1. Since Zfp36 repressed the translation of the FLuc-5BoxB-SV40pA transcript (Figure 4) but not that of the FLuc-5BoxB-MALAT1 transcript (Figure 4), we conclude that Zfp36-mediated translation is dependent on Pabpc1.

In the manuscript, we identified the Zfp36-mF mutant because we wanted to use a minimal Zfp36 fragment that can be stably expressed and interacts with Pabpc1. Thereby, when over-expressed, this Zfp36 fragment can attenuate the endogenous Zfp36:Pabpc1 interaction. The Zfp36-mF fragment suits well for this purpose. As the reviewers described, the Zfp36-mF lacks both the N- and C-termini that interact with previous characterized translational repressors (GYF2/4EHP and Ccr4-Not complex, respectively). Thus, when over-expressed, the Zfp36-mF attenuates the endogenous Zfp36:Pabpc1 interaction (Figure 6), but not the potential interactions of the endogenous Zfp36:GYF2/4EHP or Zfp36:Ccr4-Not complex. Therefore, the specific translational up-regulation of Zfp36 target mRNAs caused by Zfp36-mF over-expression is due to the disruption of the Zfp36:Pabpc1 interaction but not the potential Zfp36:GYF2/4EHP or Zfp36:Ccr4-Not complex interactions. These results support our conclusion that the Zfp36:Pabpc1 interaction is functionally important for Zfp36-mediated translational repression.

Pabpc1 promotes mRNA translation in multiple steps. Our data indicate that the translational repression mediated by Zfp36 is dependent on Pabpc1. We speculate that Zfp36 represses its target mRNA translation by compromising Pabpc1’s ability to maintain mRNA in a translation-competent state.

*The role of Zfp36 as a translational silencer (in addition to its role as an inducer of mRNA degradation) has been reported previously, through the work of the Gaestel, Brooks and Lykke-Andersen labs. The authors fail to mention any of these studies, which is, at best, careless. The authors need to quote these publications adequately and discuss their findings in the light of these previous studies.*

We apologize for this oversight. In the revised manuscript, in addition to citing review papers, we also included the papers describing the original findings from these labs.